# N-glycosylation of NANOG regulates stemness and apoptosis in colon cancer cells

**Yin Tian[1]☯, Yu Zeng[2]☯, Yun Liu[3], Jun Ye[3], Xin Zhang ⓘ[1]\***

**1** Department of Gastroenterology, Yubei District People's Hospital, Chongqing, China, **2** Department of Anesthesiology, Southwest Hospital, The First Affiliated Hospital of Army Medical University, Chongqing, China, **3** Department of Gastroenterology, Southwest Hospital, The First Affiliated Hospital of Army Medical University, Chongqing, China

☯ These authors shall share the first authorship.
\* tianyindoctor@163.com

## Abstract

To investigate the effect of N-glycosylation on NANOG regulation of colon cancer stem cell characteristics. The GEPA database was used to screen and analyze the expression of N-glycosylase in colon cancer tissues. CD133＋stem cells were selected by magnetic bead sorting of colon cancer HCT116 cells and LoVo cells. Plasmid transfection of colon cancer stem cells was performed by Lipofectamine™ 3000. Cell activity was detected by MTT method. Microsphere formation test was used to detect the diameter and number of stem cell spheres. EdU flow cytometry was used to detect cell proliferation. Scratch assay was used to detect cell migration ability. Western Blot was used to detect the expression level of apoptosis-related proteins. Compared with the control group, colon cancer stem cells transfected with mutant expression vectors with N-glycosylation site deletion had reduced cell activity, decreased proliferation and migration ability; reduced tumor stem cell sphere formation ability; and increased intracellular apoptosis level. Conclusively, The seven N-glycans in the carboxyl terminus of human NANOG are involved in the molecular quality control of NANOG protein and the maintenance of the stem cell characteristics of colon cancer stem cells, further affecting the proliferation and migration ability of colon cancer stem cells.

## 1. Introduction

Colorectal cancer (CRC) is the third most common malignant tumor in the world, and its mortality rate ranks second [1]. In the Chinese disease statistics survey, the number of new cases of colorectal cancer is increasing rapidly. Compared with 365,000 new cases in 2015, the number of new cases will reach 554,000 by 2020, an increase of 51.7% [2]. In the treatment of colorectal cancer, surgery is still an effective means of clinical cure for patients with early colorectal cancer. Due to the

**Data availability statement:** All relevant data are within the paper and its Supporting Information files.

**Funding:** This study was supported by the Chongqing Natural Science Foundation General Project[CSTB2022NSCQ-MSX0454]. The role of the funder is to design the study and make the decision to publish.

**Competing interests:** The authors have declared that no competing interests exist.

popularization of screening programs, more than 70% of new cases in 2015 underwent potentially curative resection. In fact, the five-year overall survival rate depends on the stage of the disease (90% for stage I and 15% for stage IV) [3]. Given the low five-year survival rate of patients with advanced disease, it is imperative to find targets and strategies that can improve the prognosis of patients with advanced colorectal cancer.

Induced pluripotent stem cells (iPSCs) are a type of reprogrammed cells that are transformed from terminally differentiated somatic cells into stem cells through the induction of specific transcription factors. They have developmental pluripotency similar to that of embryonic stem cells, such as the ability to self-renew and differentiate into various cell types [4]. In CRC, normal and malignant adult cells retain the ability to be reprogrammed into iPSCs. In 2013, Schwitalla et al. found that differentiated non-stem cells in the epithelium of colon tissue can be transformed into cancer stem cells (CSCs) in vivo under the action of the inflammatory factor NF-kB[5]. CSCs are believed to play a key role in the initiation, progression, metastasis and recurrence of cancer due to their ability to self-renew and form tumor masses. Embryonic stem cells (ESCs) nuclear transcription factors (TFs) OCT4, SOX2 and NANOG together constitute a key transcriptional network for pluripotency, which recruits chromatin modification proteins and collaborates with genes such as c-Myc and Klf4 to further induce broad transcriptional activation [6], thereby jointly inducing somatic cell reprogramming into pluripotent stem cells.

NANOG is a homologous chromosomal protein that is regulated at the allelic level to maintain cell pluripotency and play a role in supporting the self-renewal of embryonic stem cells. Its absence will lead to the loss of primitive endoderm development [7]. Inhibiting NANOG gene expression may reduce tumor pathogenicity and stem cell characteristics in vitro, such as anchorage-dependent growth, while overexpression may enhance tumor cell stemness expression. Studies have found that simply high expression of NANOG can induce the differentiation of tumor cells into iPSCs and regulate the self-renewal of cancer stem cells [8]. Human NANOG protein is encoded by the NONOG1 gene and contains 305 amino acids and a highly conserved DNA homology binding region. The NANOG1 protein structure can be divided into ND (amino-terminal domain), HD (homology domain), and CD (carboxyl-terminal domain). Related studies have shown that epigenetic modifications of the 5-terminal upstream promoter region of the NANOG gene Significantly affects the expression of this gene. Histone acetylation, methylation modification and arginine methylation can all affect the expression of NANOG, thereby regulating CSC self-proliferation and sub-totipotency [9,10].

N-glycosylation is a common post-translational modification of proteins in eukaryotic cells. It plays an important role in many biological regulatory processes, including immune response and protein quality control system. N-glycans are key signals for protein folding, QCS (quality control system), endoplasmic reticulum-associated degradation (ERAD) and protein transport in eukaryotic cells. Abnormal N-glycosylation leads to abnormal protein folding and is intercepted by molecular chaperones such as calreticulin and calnexin, and then degraded by ubiquitin proteasomes [11,12].

The human OST complex OST-A (containing STT3A) and OST-B (containing STT3B) are involved in the N-linked glycosylation of proteins in the endoplasmic reticulum [13]. Compared with non-transformed tumor cells, tumor cells often show extensive N-glycosylation changes, which play a crucial role in the occurrence and development of tumors. Analysis of the N-glycan spectrum of human embryonic stem cells showed that the phenotype of the N-glycan spectrum is closely related to its differentiation state. As its differentiation progresses, the phenotype of its undifferentiated N-glycans is gradually replaced by the phenotype of the N-glycans of differentiated cells, suggesting that N-glycosylation can be used as a strategy for targeted stem cell therapy. Therefore, abnormal N-glycosylation is usually an important biomarker and provides a set of specific targets for therapeutic intervention. However, whether there is N-glycosylation at the carboxyl terminus of the NANOG protein molecule and the role of N-glycosylation in regulating the stemness of CSC cells and inducing the differentiation of iPSCs remain to be elucidated.

In this study, through the analysis of NANOG amino acid sequence, we preliminarily revealed whether N-glycosylation in the carboxyl terminus of human NANOG protein is involved in the quality control of Nanog protein, and further verified to clarify whether a specific N-glycosidic sugar chain (N-glycans) site is involved in regulating the maintenance of stemness of colon cancer precursor cells.

## 2. Materials and methods

### 2.1. CRC culture and characterization

Colon cancer HCT116 cells (CL-0096) and LoVo cells (CL-0144) were purchased from Shanghai Prosperity Biotechnology Co., Ltd. The cells were cultured in DMEM high-glucose medium(12100046, Gibco, USA) containing 10% fetal bovine serum(10088–141, Gibco, USA) at 37°C and 5% $CO_2$. The medium was changed every 2–3 days, and the cells were passaged after growing to 70% cell confluence, with a passage ratio of 1:3. HCT116 cells and LoVo cells were resuspended and centrifuged at a density of $1 \times 10^7$, and buffer, FcR Blocking Reagent and CD133 magnetic beads were added in sequence according to the CD133 magnetic beads sorting kit(130-100-857, Univ, China) operation procedure, and incubated on ice in the dark. CD133+cells were then collected using LS magnetic beads.

### 2.2. Plasmids transfection

The entire human NANOG coding region or its mutants was designed and cloned in a *pcDNA3.1*+plasmid. The plasmids were extracted following the instructions of EndoFree Mini Plasmid Kit (TIANGEN BIOTECH, China), and transfected into HCT116 stem cells and LoVo stem cells with Lipofectamine 3000 (Thermo Fisher Scientific). Briefly, The obtained HCT116 stem cells and LoVo stem cells were inoculated into 6-well plates, and 2 μg of Nanog WT or MUT (MUT-1, MUT-2, MUT-3, MUT-4, MUT-5, MUT-6, MUT-7) plasmids were added to 500 μL of serum-free medium and gently mixed; 5 μL of Lipofectamine™ 3000 was added to 500 μL of serum-free medium and gently mixed, and placed at room temperature for 5 minutes; the diluted Nanog plasmid was mixed with Lipofectamine™ 3000, gently mixed, and placed at room temperature for 20 min. The Nanog plasmid/Lipofectamine™ 3000 complex was added to the wells of the culture plate containing cells, and the cell culture plate was gently shaken back and forth; after incubation for 4–6 hours, the complex was removed and replaced with 10% FBS tumor stem cell complete medium according to the above grouping, and cultured in a 37°C, 5% CO2 incubator for 48 hours.

### 2.3. MTT assay for cell proliferation

HCT116 stem cells and LoVo stem cells were inoculated into 96-well culture plates at a density of $1 \times 10^5$/ml, 100 μl per well. After plasmid transfection of cells according to the aforementioned cell transfection steps, the culture medium in the culture plate was aspirated, 100 μl of freshly prepared 5 mg/ml MTT solution was added to each well, and incubated for 4 hours. The supernatant was aspirated again and 200 μl of DMSO solution was added to each well, and the optical density (OD) at a wavelength of 490 nm was measured using a multifunctional microplate reader.

## 2.4. EDU flow cytometry for cell proliferation

After plasmid cell transfection according to the aforementioned experimental steps, HCT116 stem cells and LoVo stem cells were prepared into single cell suspensions at a density of $1 \times 10^7$/ml, and cell proliferation was detected on a flow cytometer according to the instructions of the EDU staining kit(C0075L, Beyotime, China). Briefly, The preheated EdU working solution was added into cells and cultured for 2hours, and then cells were fixed with 4% paraformaldehyde. After washed three times with PBS, cells were incubated with 0.3% Triton X-100 in PBS for 15 min, and Click Additive Solution was added for 30 min. After removing the Click reaction solution and washed, cells were detected on a flow cytometer.

## 2.5. Tumor stem cell sphere formation experiment

HCT116 stem cells and LoVo stem cells were seeded into 6-well plates at a density of $1 \times 10^5$/ml respectively. After plasmid cell transfection according to the aforementioned experimental steps, the cell culture plates were gently shaken and the culture medium was replaced with serum-free tumor stem cell complete culture medium. After culturing in a 37°C, 5% CO2 incubator for 48 h, the plates were photographed under a microscope and the sphere diameter was measured using WZ_Camera software.

## 2.6. Scratch assay to detect cell migration

The obtained cells were prepared into a single cell suspension, and the cell density was adjusted to $5 \times 10^5$/ml and inoculated into a 6-well culture plate. After the cells were fully covered, vertical scratches were made and transfection was performed according to the above experimental groups. After incubation for 4–6 hours, the culture medium was replaced with complete tumor stem cell culture medium containing 10% FBS. Samples were taken at 0h and 24h and photos were taken. The migration distance was measured using WZ_Camera software.

## 2.7. Western blot assay

After removing the cell culture medium and washing the cells with PBS, total cellular proteins were harvested using a protein lysis buffer supplemented with PMSF and protease inhibitor cocktail at the recommended ratios. The cell protein samples were subjected to ultrasonic lysis, followed by centrifugation to collect the supernatant. Protein concentrations were determined using the BCA assay. Subsequently, the protein samples were diluted with SDS-PAGE loading buffer at the appropriate ratio, boiled, and then subjected to loading, detection, and development according to standard Western blot protocols. The primary antibodies used in the experiments were as follows: Nanog (Bioss, cat. no. bsm-51459M, 1:1000 dilution), Bcl-2 (Bioss, cat. no. bsm-52022M, 1:500 dilution), Bax (Abcam, cat. no. ab32503, 1:1000 dilution), and GAPDH (Bioss, cat. no. bsm-33033M, 1:10000 dilution).

## 2.8. Statistical analysis

SPSS 22.0 statistical software and Graphpad Prizm8 were used for statistical analysis. Independent sample t-test was used for comparison between two groups, and one-way analysis of variance was used for comparison between multiple groups. $P < 0.05$ was considered statistically significant.

## 3. Results

### 3.1. Abnormal N-glycosylation can predict poor prognosis of CRC patients

To evaluate the clinical significance of NANOG upregulation in colon cancer, we analyzed the expression of N-glycosylases in colon cancer tissues from the Gene Expression Profiling Interactive Analysis (GEPA) database. The results showed that the expression of human OST complexes OST-A (containing STT3A) and OST-B (containing STT3B), which regulate N-glycosylation, was significantly higher in colon cancer tissues than in normal colon tissues (Fig 1).

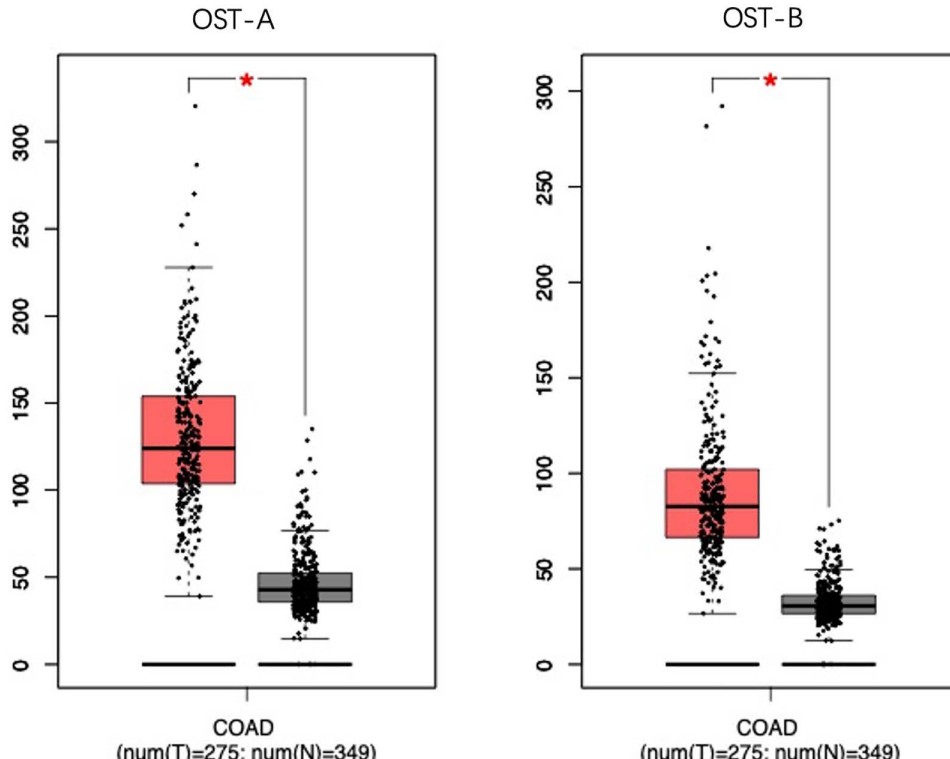

**Fig 1. GEPA database shows that upstream target genes regulating N-glycosylase are highly expressed.**

### 3.2. NANOG is N-glycosylated at the NXT motif site in CRC stem cells

To identify the N-glycosylation sites of NANOG in CRC cells, we searched for the evolutionarily conserved NXT motif in the amino acid sequence of human NANOG. According to our analysis of the 305 amino acid sequences of human NANOG protein, we found that there was one N-glycosylation site in the HD domain and six enriched N-glycosylation sites in its carboxyl terminal domain (Fig 2A). To further verify whether the seven N-glycosidic sugar chains (N-glycans) in the base segment of the human NANOG gene are involved in the molecular quality control of NANOG protein, we cloned and expressed the NANOG protein encoding 305 amino acids, and constructed seven mutant expression vectors corresponding to the deletion of N-glycosylation sites, in which the N-X-T (S) encoding the N-glycosylation site was mutated to A-X-T (S), and then constructed the NANOG-WT cell model in CD133 + cells in HCT116 and LoVo colon cancer cells. WB experiments showed that since the molecular weight of each N-glycoside sugar chain is 2 kDa, after the mutant site deletion, the molecular weight of the seven NANOG-MUTs was observed to decrease to 35 kDa, indicating that N-glycosylation existed at all seven potential sites (Fig 2B).

### 3.3. N-glycosylation affects NANOG protein in regulating colon cancer stem cell proliferation

After magnetic bead sorting of CD133 + stem cells in HCT116 colon cancer cells and CD133 + stem cells in LoVo colon cancer cells, the WT plasmid and seven mutant plasmids of Nanog protein were transfected respectively. MTT assay was used to detect the effects of seven different N-glycosidic sugar mutants at the carboxyl terminus of NANOG protein on the proliferation of colon cancer stem cells. Experimental results showed that in HCT116 colon cancer stem cells and LoVo colon cancer stem cells, compared with the WT group, the seven MUT plasmid transfection groups of NANOG protein

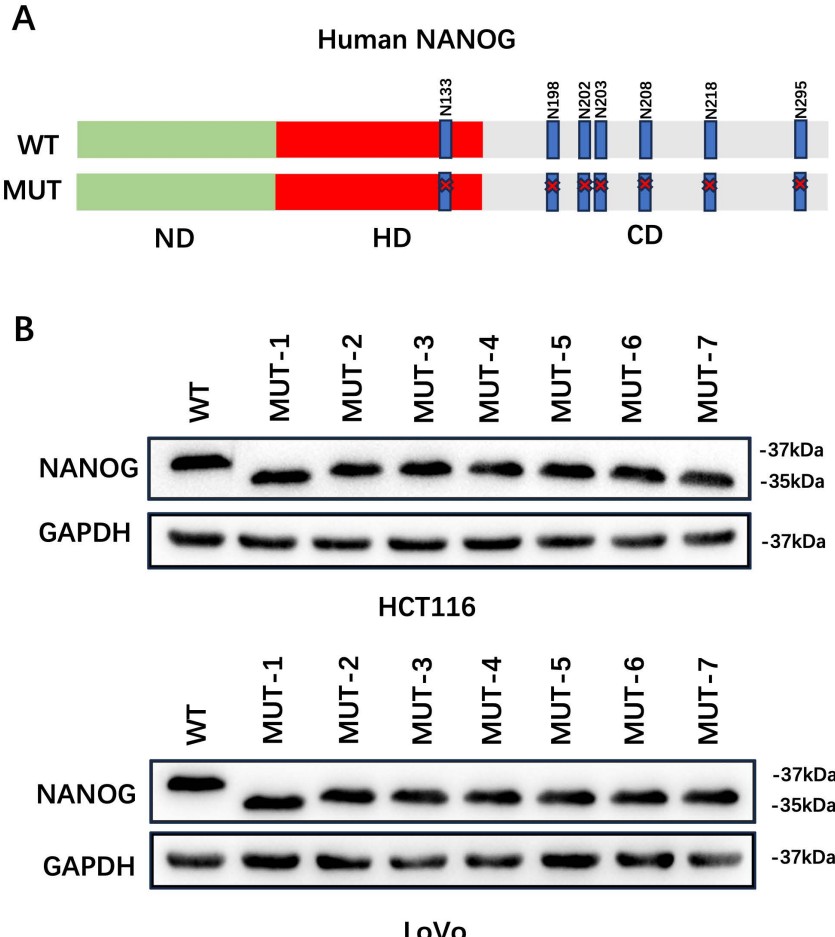

**Fig 2. NANOG has glycosylation sites. (A)** Schematic diagram of the construction of a mutant expression vector with N-glycosylation site deletion. **(B)** Western Blot shows the change in the molecular weight of NANOG caused by the MUT mutant.

could significantly inhibit the proliferation of colon cancer stem cells (Fig 3). Flow cytometry was used to detect cell proliferation. The experimental results showed that the seven MUT plasmid transfection groups could inhibit the proliferation of tumor stem cells in both groups to varying degrees. (Figs 4 and 5) These results indicate that the seven N-glycosidic sugar chains in the carboxyl terminus of the NANOG gene are involved in regulating the proliferation of colon cancer stem cells.

### 3.4. N-glycosylation affects NANOG protein in maintaining the stem cell characteristics of colon cancer stem cells

After magnetic bead sorting of CD133+stem cells in HCT116 colon cancer cells and CD133+stem cells in LoVo colon cancer cells, they were transfected with WT plasmids and 7 mutant plasmids of NANOG protein, respectively, and cultured for 72 hours. The results of the 3D stem cell sphere experiment showed that compared with the WT group, the stem cell sphere diameter and number of HCT116 colon cancer stem cells and LoVo colon cancer stem cells transfected with 7 MUT plasmids of NANOG protein were reduced (Fig 6), indicating that N-glycosylation is involved in the maintenance of stem cell characteristics by the NANOG gene.

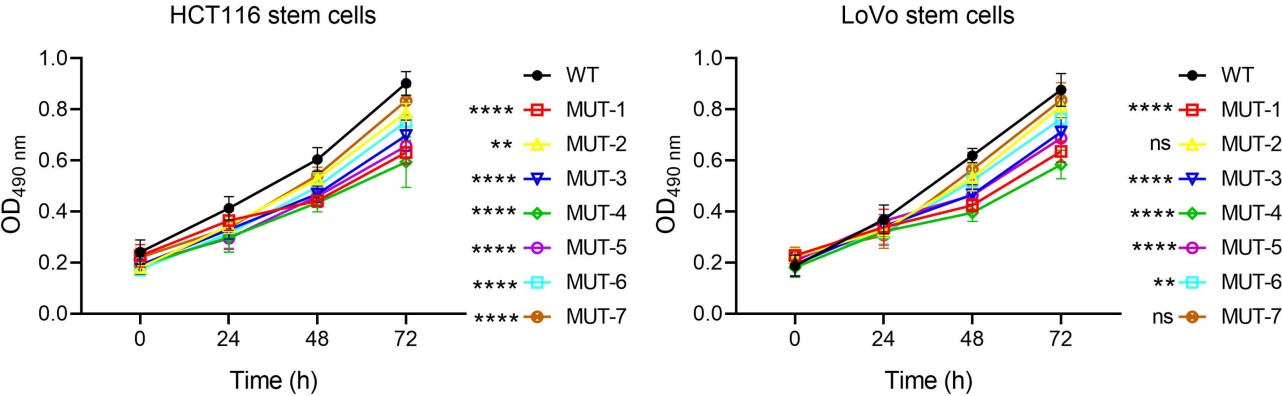

**Fig 3. MTT experiment shows that the seven MUT plasmids of NANOG protein can significantly inhibit the proliferation of colon cancer stem cells.** Compared with the control group, **p<0.01; ****p<0.0001; ns: no significant difference.

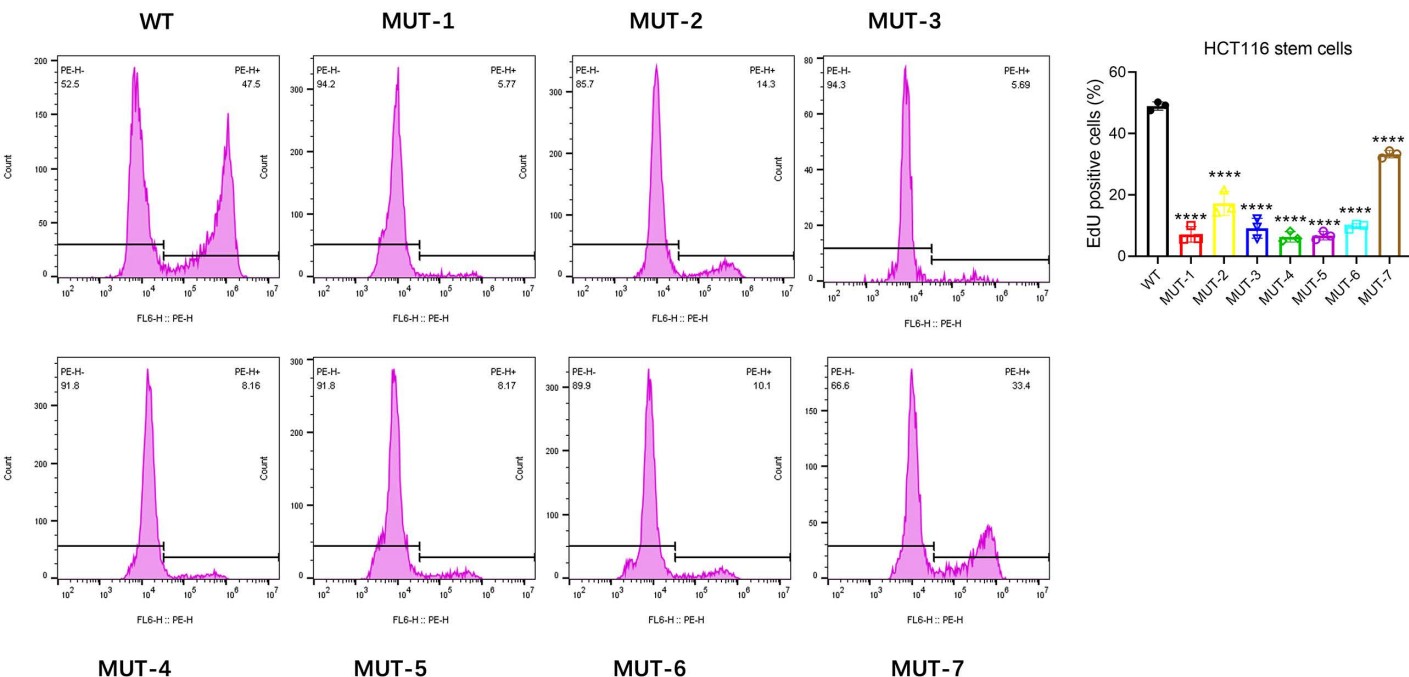

**Fig 4. EdU flow cytometry shows that the seven MUT plasmids of NANOG protein can inhibit the proliferation of HCT116 stem cells to varying degrees.** Compared with the control group, ****p<0.0001; ns: no significant difference.

### 3.5. N-glycosylation affects NANOG protein in regulating the migration ability of colon cancer stem cells

The WT plasmid and 7 mutant plasmids of NANOG protein were transfected into CD133+ stem cells in magnetic bead-sorted HCT116 colon cancer cells and cultured for 24 hours. Cell migration experiments showed that compared with the WT group, the 7 MUT plasmids of Nanog protein could inhibit the migration ability of HCT116 stem cells to varying degrees (Fig 7).

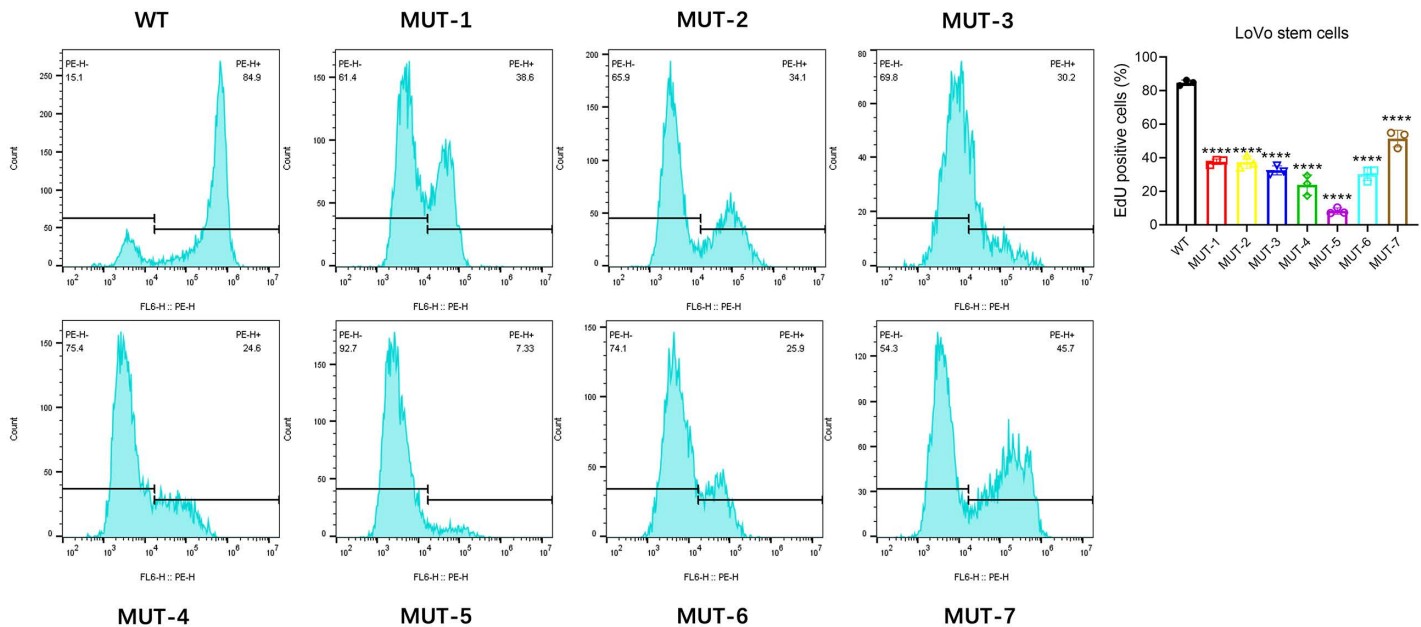

**Fig 5. EdU flow cytometry shows that the seven MUT plasmids of Nanog protein can inhibit the proliferation of LoVo stem cells to varying degrees.** Compared with the control group, ****p < 0.0001; ns: no significant difference.

### 3.6. N-glycosylation enhances NANOG-mediated apoptosis of colon cancer stem cells

After magnetic bead sorting of CD133 + stem cells in HCT116 colon cancer cells and CD133 + stem cells in LoVo colon cancer cells, the WT plasmid and 7 mutant plasmids of NANOG protein were transfected respectively and cultured for 72 hours. WB experiments detected apoptosis-related proteins Bax and bcl2. The results showed that the apoptosis levels of HCT116 colon cancer stem cells and LoVo colon cancer stem cells after transfection of 7 MUT plasmids of NANOG protein decreased (Fig 8), indicating that N-glycosylation enhances NANOG-mediated apoptosis of tumor stem cells.

## 4. Discussion

Mounting evidence suggests that aberrant glycosylation patterns of glycoproteins, such as the presence of multi-branched N-glycans, are closely linked to the malignant transformation and metastatic behavior of cancer cells [14,15]. Based on the type of amino acid residues they attach to, protein glycosylation is mainly classified into two categories. Among them, N-glycosylation refers to the covalent linkage of carbohydrates to asparagine (Asn) residues, forming the typical consensus sequence N-X-S/T (X ≠ P) [16]. N-linked glycosylation is a common post-translational modification, formed by the covalent attachment of oligosaccharides to asparagine residues in polypeptide chains, and plays an important role in determining the folding state and oligomerization of proteins. Aberrant N-glycosylation leads to abnormal protein folding, which is intercepted by molecular chaperones such as calreticulin and calnexin, and subsequently degraded by the proteasome [12,17]. Studies on the autocatalytic cleavage process of the conserved SEA module in the carboxyl terminus of rat mucin Muc3 have revealed that the seven densely distributed N-glycans within the SEA module play a key role in the autocatalytic cleavage occurring within the enzymatic cleavage motif LSKGSIVV [18–20]. In the NANOG protein, we observed that it contained one N-glycosylation site in its HD domain and six enriched N-glycosylation sites in its carboxyl terminal domain. Further, by comparing the differences in its molecular weight in eukaryotic cells, it was confirmed that all seven potential sites were N-glycosylated.

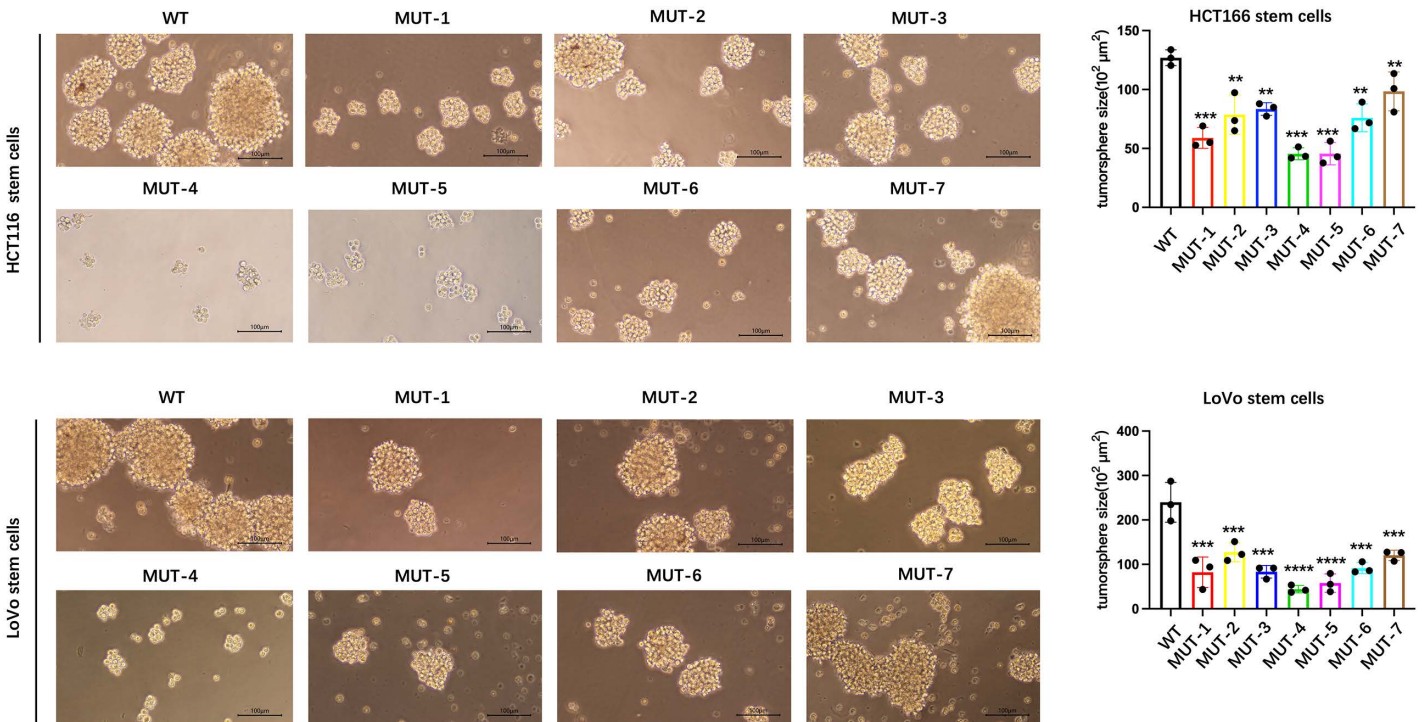

**Fig 6. Stem cell sphere formation test shows that compared with the WT group, the seven MUT plasmids of NANOG protein can inhibit the stem cell sphere formation efficiency of HCT116 stem cells and LoVo stem cells.** Scale bar 100μm. Compared with the control group, **p < 0.01; ***p p < 0.001; ****p < 0.0001;.

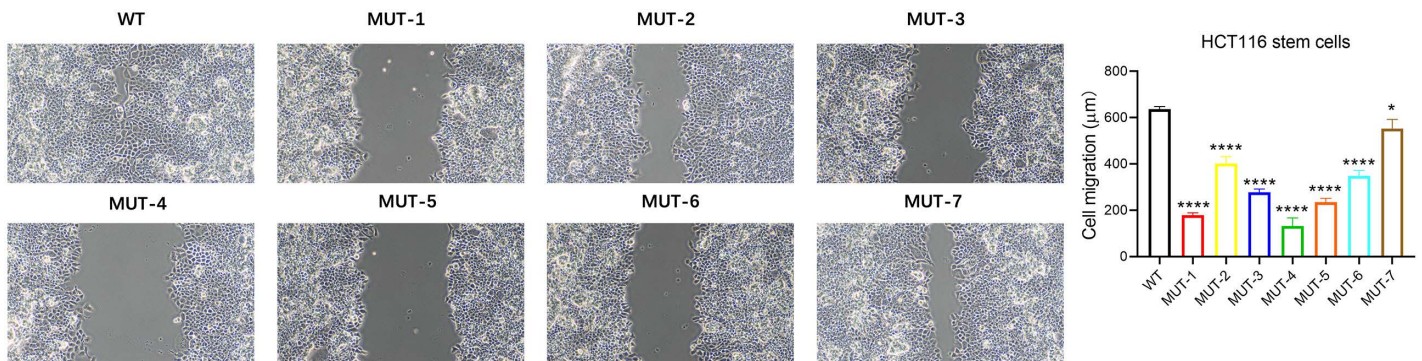

**Fig 7. Cell migration assay shows that compared with the WT group, the seven MUT plasmids of NANOG protein can inhibit the migration ability of HCT116 stem cells.** *p < 0.05; ****p < 0.0001.

NANOG is a transcription factor in embryonic stem cells and hematopoietic stem cells that plays a central role in maintaining stem cell self-renewal and pluripotency. NANOG regulates various aspects of cancer development, such as tumor cell proliferation, migration, EMT, immune evasion, drug resistance, malignant transformation, and communication between cancer cells and the surrounding stroma [21]. Human NANOG encodes a 305-amino acid protein, which includes the N-terminal (amino acids 1–94), the DNA-binding homeodomain (amino acids 95–154), and the C-terminal

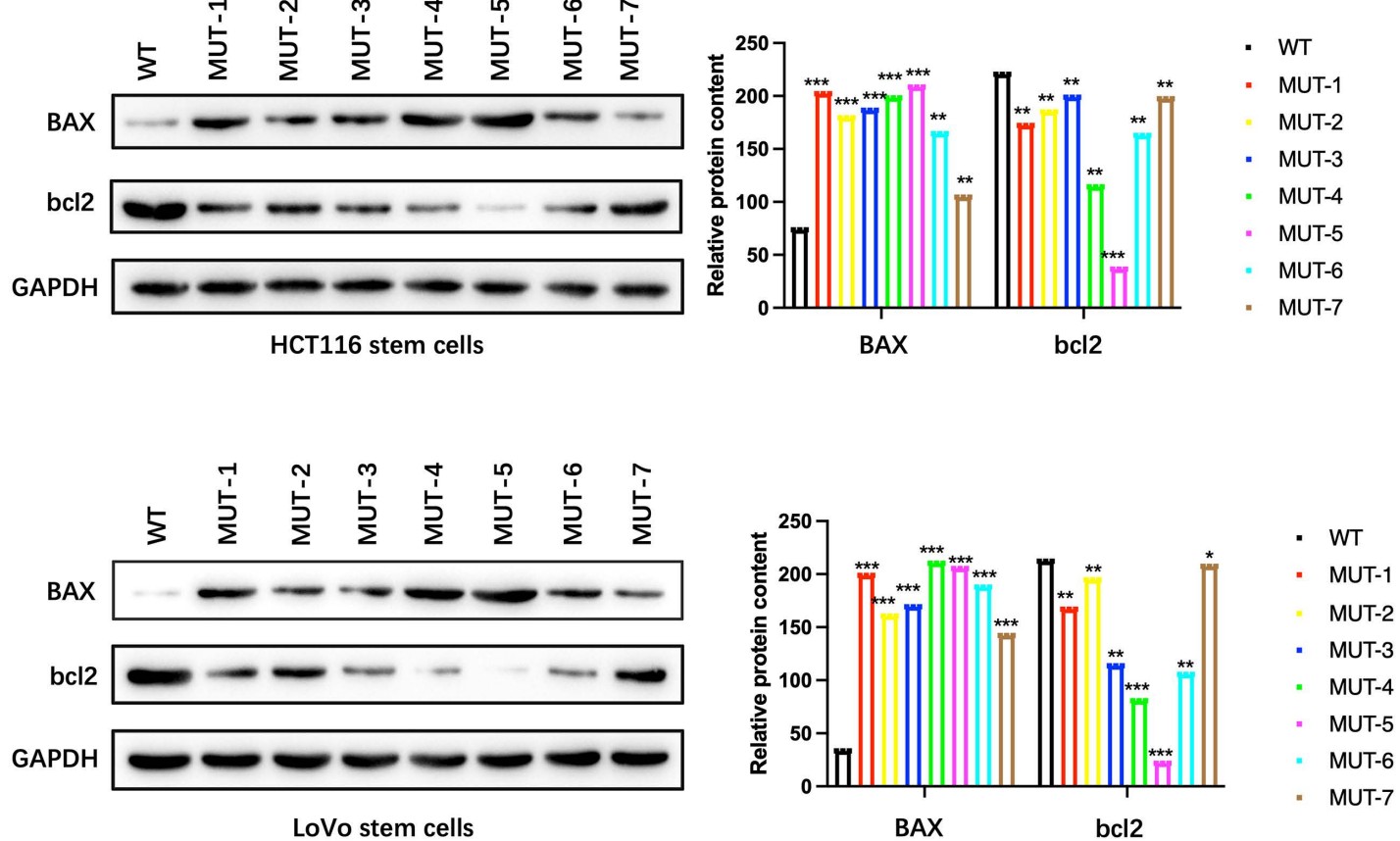

**Fig 8. WB experiment shows the changes in BAX and bcl2 protein expression levels.** Compared with the control group, **p < 0.01; ***p p < 0.001;.

(amino acids 155–305) domains. The C-terminal domain consists of three regions, namely the upstream (amino acids 155–195), the tryptophan-rich domain (amino acids 196–240), and the downstream region (amino acids 241–305) [22]. In this study, a mutant expression vector with 7 N-glycosylation sites deleted was constructed, the N-X-T(S) encoding the N-glycosylation site was mutated to A-X-T(S), and then transfected into colon cancer stem cells. We found that all 7 N-glycosylation sites are involved in the regulation of the stem cell properties of colon cancer stem cells by NANOG protein. Understanding NANOG gene expression and its regulation is a key step in understanding early embryogenesis, pluripotent cell development, and cancer cell proliferation. The above studies have confirmed that the carboxyl terminus of the NANOG protein molecule is N-glycosylated, and N-glycosylation has a certain regulatory effect on the stemness of tumor cells.

## 5. Conclusion

This study revealed that 7 N-glycosidic sugar chains (N-glycans) in the carboxyl terminus of human NANOG are involved in the molecular quality control of NANOG protein and the maintenance of stem cell characteristics of colon cancer stem cells, further affecting the proliferation and migration ability of colon cancer stem cells. Subsequent studies may verify the effect of N-glycosylation on the tumorigenicity of NANOG gene through in vivo experiments to further verify the role of N-glycosylation in regulating cell stemness and inducing iPS phenomenon.

## Supporting information

**S1 File. The base sequences of the WT plasmid and seven mutant plasmids of Nanog protein.**
(DOCX)

**S2 File. Raw data processing for statistical charts.**
(DOCX)

**S3 File. Original Western Blot images.**
(PDF)

## Author contributions

**Conceptualization:** Yin Tian.

**Data curation:** Yu Zeng, Yun Liu.

**Funding acquisition:** Xin Zhang.

**Investigation:** Yin Tian.

**Methodology:** Yu Zeng.

**Writing – original draft:** Yin Tian, Xin Zhang.

**Writing – review & editing:** Yun Liu, Jun Ye, Xin Zhang.

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
