## [Editor Report · Decision Letter 0]

29 May 2025

Dear Dr. zhang,

Thank you for submitting your manuscript to PLOS ONE. After careful consideration, we feel that it has merit but does not fully meet PLOS ONE’s publication criteria as it currently stands. Therefore, we invite you to submit a revised version of the manuscript that addresses the points raised during the review process.

We look forward to receiving your revised manuscript.

Kind regards,

Gianpaolo Papaccio, M.D., Ph.D.

Academic Editor

PLOS ONE

10.1016/j.apsb.2023.10.014

https://doi.org/10.1002/jcp.30063

https://doi.org/10.1016/j.apsb.2023.10.014

In your revision ensure you cite all your sources (including your own works), and quote or rephrase any duplicated text outside the methods section. Further consideration is dependent on these concerns being addressed.

“This study was supported by the Chongqing Natural Science Foundation General Project[CSTB2022NSCQ-MSX0454]”

“This study was supported by the Chongqing Natural Science Foundation General Project[CSTB2022NSCQ-MSX0454].”

“This study was supported by the Chongqing Natural Science Foundation General Project[CSTB2022NSCQ-MSX0454]”

7. Please include a separate caption for each figure in your manuscript.

Additional Editor Comments:

This manuscript investigates the impact of N-glycosylation on NANOG protein and its role in regulating colon cancer stem cell characteristics, proliferation, migration, and apoptosis. The study addresses a relevant topic and the experimental design seems sound in principle. The findings suggest a novel regulatory mechanism for NANOG, a key pluripotency factor in cancer. While the study presents interesting data, the manuscript requires significant revisions to improve clarity, methodological detail, and discussion of limitations. Crucially, the absence of figures makes a comprehensive assessment of the results impossible.

Major Comments:

Title: The current title, "N-glycosylation regulates NANOG affects stem cell characteristics and apoptosis of colon cancer cells," is grammatically awkward, lengthy, and unclear. It should be revised for precision and conciseness.

Missing Figures (Very Critical): All figures (Figure 1 to Figure 8) mentioned throughout the manuscript are absent. Without the visual representation of the data, it is impossible to properly evaluate the results, the statistical significance claims, and the interpretations drawn by the authors. This is a very critical omission. For instance, the verification of molecular weight shifts in NANOG-MUTs (Figure 2B), the quantification of cell activity, proliferation, migration, sphere formation, and apoptosis (Figures 3-8) cannot be assessed.

There is an inconsistency in the manuscript regarding the HCT166/HCT116 cell line. The abstract refers to HCT166, while the Materials and Methods section initially lists HCT116 (CL-0096) and LoVo (CL-0144) but later refers to HCT166 cells in the same section. This must be clarified and consistent throughout the manuscript.

There is a lack of detailed Western Blot Protocol. While Western Blot experiments are mentioned as a method to detect apoptosis-related proteins and verify protein shifts, a dedicated "Western Blot" subsection outlining the detailed protocol (e.g., antibodies used, dilutions, blocking conditions, detection method) is missing from the Materials and Methods. This is essential for reproducibility.

Apoptosis Results Description is ambiguous: In Section 3.6, the statement "the apoptosis levels of HCT116 colon cancer stem cells and LoVo colon cancer stem cells after transfection of 7 MUT plasmids of NANOG protein decreased (Figure 8)" needs careful rephrasing for clarity. If N-glycosylation is stated to "promote apoptosis" (as in the abstract and section title), then the mutation/removal of glycosylation sites (via MUT plasmids) should lead to decreased apoptosis, which aligns with the finding. However, the phrasing "N-glycosylation participates in regulating NANOG protein to promote apoptosis" and then showing decreased apoptosis in mutants can be confusing. Consider phrasing like "N-glycosylation is required for NANOG protein to promote apoptosis" or "N-glycosylation enhances NANOG-mediated apoptosis."

Minor points:

The introduction mentions screening for "N-glycosylase" expression but could benefit from explicitly naming OST-A and OST-B earlier, as these are the specific enzymes identified and discussed in the results.

While the general mutation strategy is stated (N-X-T(S) to A-X-T(S)), more specific information about the exact mutation sites for each of the seven mutants (MUT-1 to MUT-7) would significantly enhance the reproducibility of the study.

Lack of Rationale for CD133+ Cell Selection: why CD133+ cells were selected as the primary stem cell population for colon cancer or provide a relevant citation to support this choice.

Provide additional details or a reference for the "WZ_Camera" software used for measurements, if it is not a widely recognized commercial software.

To address the novelty, the authors should explicitly articulate the unique knowledge gap addressed by their study in the introduction and ensure the discussion clearly differentiates their findings from existing literature on NANOG or N-glycosylation in cancer.

---

## [Author Response · Author response to Decision Letter 1]

14 Aug 2025

Response: Thanks for the suggestions. we misunderstood the format requirements and we’ve revised the manuscript format to meet PLOS ONE’s style requirements as possible.

2. We noticed you have some minor occurrence of overlapping text with the following previous publication(s).

Response: Thanks for the suggestions. During the writing of the manuscript, some content cited from references was missing proper citations. In the revised version, we have rephrased some of the text and supplemented the citations of the references.

3. Please state what role the funders took in the study.

Response: Thanks for the suggestions. The role of the funder is to design the study and make the decision to publish. We may omitted this information in the online submission system.

4. Please remove any funding-related text from the manuscript and let us know how you would like to update your Funding Statement.

Response: Thanks for the suggestions. We removed the funding-related text from the manuscript, and it need no more update.

5. Please ensure that you have an ORCID iD and that it is validated in Editorial Manager.

Response: Thanks for the suggestions. We validated the ORCID in Editorial Manager.

6. In your cover letter, please note whether your blot/gel image data are in Supporting Information or posted at a public data repository.

Response: Thanks for the suggestions. We attached the original blot/gel image data in the Supporting Information file, which named WB original pictures, and uploaded with other files in the online submission system.

7. Please include a separate caption for each figure in your manuscript.

Response: Thanks for the suggestions. During the submission process, we may have misunderstood the submission requirements and uploaded the figures along with their corresponding textual descriptions as a separate file (named "Figures") to the online submission system. Another editor also pointed out the issue of missing figures and tables. In this major revision, we will incorporate the key figures/tables along with their corresponding descriptions into the “revised manuscript with track changes”. If this still does not meet the submission format requirements, please inform us promptly so that we can make further corrections.

Major Comments:

1. The current title, "N-glycosylation regulates NANOG affects stem cell characteristics and apoptosis of colon cancer cells," is grammatically awkward, lengthy, and unclear. It should be revised for precision and conciseness.

Response: Thanks for the suggestions. We’ve changed the title as “N-glycosylation of NANOG regulates stemness and apoptosis in colon cancer cells”due to the comments.

2. Missing Figures (Very Critical) problem.

Response: Thanks for the suggestions. During the submission process, we may have misunderstood the submission requirements and uploaded the figures along with their corresponding textual descriptions as a separate file (named "Figures") to the online submission system. In this major revision, we will incorporate the key figures/tables along with their corresponding descriptions into the “revised manuscript with track changes”. If this still does not meet the submission format requirements, please inform us promptly so that we can make further corrections.

3. There is an inconsistency in the manuscript regarding the HCT166/HCT116 cell line.

Response: Thanks for the suggestions. In this study, In this study, two colon cancer cell lines, HCT116 and LoVo colon cancer cells, were utilized for the relevant experiments. We sincerely apologize for mistakenly writing HCT116 as HCT166 in some passages. We have corrected this error at the corresponding positions.

4. There is a lack of detailed Western Blot Protocol.

Response: Thanks for the suggestions. We sincerely apologize that during the manuscript writing process, we omitted the operational procedures for the Western blot (WB) experiments, which have now been supplemented in the revised manuscript.

5. Apoptosis Results Description is ambiguous: In Section 3.6, the statement "the apoptosis levels of HCT116 colon cancer stem cells and LoVo colon cancer stem cells after transfection of 7 MUT plasmids of NANOG protein decreased (Figure 8)" needs careful rephrasing for clarity. If N-glycosylation is stated to "promote apoptosis" (as in the abstract and section title), then the mutation/removal of glycosylation sites (via MUT plasmids) should lead to decreased apoptosis, which aligns with the finding. However, the phrasing "N-glycosylation participates in regulating NANOG protein to promote apoptosis" and then showing decreased apoptosis in mutants can be confusing. Consider phrasing like "N-glycosylation is required for NANOG protein to promote apoptosis" or "N-glycosylation enhances NANOG-mediated apoptosis."

Response: Thanks for the suggestions. The mutation/removal of glycosylation sites was achieved via transfection of mutant plasmids. Our findings revealed that the elimination of NANOG glycosylation led to a reduction in the apoptosis level of cancer stem cells. We appreciate the editor's comments and have revised the subheading to convey the information more accurately.

Minor points:

1. The introduction mentions screening for "N-glycosylase" expression but could benefit from explicitly naming OST-A and OST-B earlier, as these are the specific enzymes identified and discussed in the results.

Response: Thanks for the suggestions. In the Introduction section, we first introduce the NANOG protein, followed by a discussion on the impact of aberrant N-glycosylation on NANOG-mediated regulation of cancer stem cells. Subsequently, we cite literature reports indicating that human OST complexes, OST-A (containing STT3A) and OST-B (containing STT3B), are involved in N-linked glycosylation of proteins in the endoplasmic reticulum. We believe that explicitly mentioning OST-A and OST-B at an earlier stage may slightly compromise the current logical flow and coherence of the manuscript. We welcome any further detailed suggestions from the editor and are happy to make revisions accordingly.

2. While the general mutation strategy is stated (N-X-T(S) to A-X-T(S)), more specific information about the exact mutation sites for each of the seven mutants (MUT-1 to MUT-7) would significantly enhance the reproducibility of the study.

Response: Thanks for the suggestions. We will include additional information on the 7 mutants in the supplementary files, including the exacttarget sequence of each mutant.

3. Lack of Rationale for CD133+ Cell Selection: why CD133+ cells were selected as the primary stem cell population for colon cancer or provide a relevant citation to support this choice. Response: Thanks for the suggestions. CD133 (prominin-1) was selected as a marker for isolating colon cancer stem cells (CSCs) based on extensive prior evidence demonstrating its association with stemness properties in colorectal cancer. Multiple studies have shown that CD133+ subpopulations in colon cancer exhibit enhanced self-renewal capacity, tumorigenicity in xenograft models, and resistance to chemotherapy—hallmarks of CSCs.( doi:10.3390/ijms24032400)

4. Provide additional details or a reference for the "WZ_Camera" software used for measurements, if it is not a widely recognized commercial software.

Response: Thanks for the suggestions. WZ_Camera is a domestically developed image measurement software with a more concise and user-friendly interface compared to widely recognized commercial software. The images in the manuscript were also analyzed using Image J, and the resulting data were consistent.

---

## [Editor Report · Decision Letter 1]

27 Aug 2025

Dear Dr. zhang,

Thank you for submitting your manuscript to PLOS ONE. After careful consideration, we feel that it has merit but does not fully meet PLOS ONE’s publication criteria as it currently stands. Therefore, we invite you to submit a revised version of the manuscript that addresses the points raised during the review process.

We look forward to receiving your revised manuscript.

Kind regards,

Gianpaolo Papaccio, M.D., Ph.D.

Academic Editor

PLOS ONE

Journal Requirements:

Additional Editor Comments:

The Authors have answered to some critical points but not to a specific one, namely:

3.Lack of Rationale for CD133+ Cell Selection: why CD133+ cells were selected as the

primary stem cell population for colon cancer or provide a relevant citation to support

this choice. Response: Thanks for the suggestions. CD133 (prominin-1) was selected

as a marker for isolating colon cancer stem cells (CSCs) based on extensive prior

evidence demonstrating its association with stemness properties in colorectal cancer.

Multiple studies have shown that CD133+ subpopulations in colon cancer exhibit

enhanced self-renewal capacity, tumorigenicity in xenograft models, and resistance to

chemotherapy—hallmarks of CSCs.( doi:10.3390/ijms24032400; DOI: 10.1186/s13046-025-03308-8 and DOI: 10.1096/fj.12-218222).

This point needs a specific answer as well as that the Authors add citations.

Other point:

The Authors when answering to the critical points write "suggestion" often it is NOT a suggestion BUT a Major point that is critical

---

## [Author Response · Author response to Decision Letter 2]

27 Oct 2025

In this revision, we have provided additional clarification regarding a single specific issue raised by the editor and have marked the supplementary response in the "Response to Reviewers" document as follows:

3. Lack of Rationale for CD133+ Cell Selection: why CD133+ cells were selected as the primary stem cell population for colon cancer or provide a relevant citation to support this choice. Response: Thanks for your question. CD133 (prominin-1) was selected as a marker for isolating colon cancer stem cells (CSCs) based on extensive prior evidence demonstrating its association with stemness properties in colorectal cancer. In this article(doi:10.1517/14728222.2012.667404), the researcher demonstrated that CD133⁺ cells isolated from colon cancer specimens (accounting for 2.5% of total cells) consistently formed xenograft tumors in immunodeficient mice. Moreover, with serial in vivo passages, CD133⁺ cells exhibited progressively enhanced aggressiveness (manifested as accelerated tumor growth and an increased proportion of CD133⁺ cells in subsequent generations of tumors), whereas CD133⁻ cells lacked this capability. The core viewpoint of another literature(doi:10.4252/wjsc.v11.i11.920) indicates that stemness in colon cancer is "a reversible epigenetic state" rather than a fixed genetic identity, and the expression of CD133 is precisely regulated by canonical epigenetic mechanisms, which aligns with the "plasticity" characteristic of stem cells. Besides, it also pointed out that the standalone use of CD133 for identifying colorectal cancer stem cells (CR-CSCs) has limitations, but its combination with other markers (CD44, CD166) significantly improves specificity, with CD133 serving as the core "anchor marker" in such combinatorial strategies. Multiple studies have shown that CD133+ subpopulations in colon cancer exhibit enhanced self-renewal capacity, tumorigenicity in xenograft models, and resistance to chemotherapy—hallmarks of CSCs.

---

## [Decision Letter · Decision Letter 2]

30 Oct 2025

N-glycosylation of NANOG regulates stemness and apoptosis in colon cancer cells

PONE-D-25-28463R2

Dear Dr. zhang,

We’re pleased to inform you that your manuscript has been judged scientifically suitable for publication and will be formally accepted for publication once it meets all outstanding technical requirements.

Kind regards,

Gianpaolo Papaccio, M.D., Ph.D.

Academic Editor

PLOS ONE

Additional Editor Comments (optional):

Reviewers' comments:

Reviewer's Responses to Questions

**Comments to the Author**

Reviewer #1: All comments have been addressed

Reviewer #2: All comments have been addressed

2. Is the manuscript technically sound, and do the data support the conclusions?

Reviewer #1: Yes

Reviewer #2: Yes

3. Has the statistical analysis been performed appropriately and rigorously?

Reviewer #1: Yes

Reviewer #2: Yes

4. Have the authors made all data underlying the findings in their manuscript fully available?

Reviewer #1: Yes

Reviewer #2: Yes

5. Is the manuscript presented in an intelligible fashion and written in standard English?

Reviewer #1: Yes

Reviewer #2: Yes

Reviewer #1: The authors have convincingly demonstrated that seven N-glycosidic sugar chains (N-glycans) located at the carboxyl terminus of human NANOG play a crucial role in the molecular quality control of the NANOG protein and in maintaining the stem cell properties of colon cancer stem cells. These post-translational modifications further influence the proliferation and migration capacities of colon cancer stem cells.

Overall, this is an interesting and well-executed study that advances our understanding of how N-glycosylation regulates NANOG function, cellular stemness, and potentially tumorigenicity.

The manuscript has been significantly improved following the revisions. The authors have addressed all previous comments in a clear and comprehensive manner, which has considerably strengthened the quality and clarity of the work. The study is well designed, and the experiments have been conducted in an exemplary and rigorous way. The methodology is appropriate, and the results are clearly presented and convincingly discussed. The figures and data are well organized and effectively support the authors’ conclusions.

Overall, this is a well-executed and scientifically sound study that makes a valuable contribution to the field and the revisions have enhanced the manuscript substantially.

Reviewer #2: The aim of the study “N-glycosylation of NANOG regulates stemness and apoptosis in colon cancer cells” is to investigate the functional role of N-glycosylation in the regulation of NANOG protein activity, particularly in maintaining the stemness characteristics and apoptosis of colon cancer stem cells. By identifying and mutating the seven potential N-glycosylation sites at the carboxyl terminus of human NANOG, the study sought to determine how these modifications influence NANOG’s molecular stability, and consequently, the proliferation, migration, and survival of colon cancer stem cells.

The experimental approaches, including molecular cloning, CD133+ stem cell sorting, and functional assays (proliferation, migration, apoptosis), are appropriate and thoroughly performed. The data are coherently presented and adequately support the authors’ conclusions and all the required experiments have been correctly conducted and the revisions effectively addressed the previous reviewers’ concerns.

**Do you want your identity to be public for this peer review?** For information about this choice, including consent withdrawal, please see our Privacy Policy

Reviewer #1: No

Reviewer #2: No

---

## [Editor Report · Acceptance letter]

PONE-D-25-28463R2

PLOS ONE

Dear Dr. zhang,

I'm pleased to inform you that your manuscript has been deemed suitable for publication in PLOS ONE. Congratulations! Your manuscript is now being handed over to our production team.

Kind regards,

on behalf of

Prof. Gianpaolo Papaccio

Academic Editor

PLOS ONE